# Secretory Nanoparticles of *Neospora caninum* Profilin-Fused with the Transmembrane Domain of GP64 from Silkworm Hemolymph

**DOI:** 10.3390/nano9040593

**Published:** 2019-04-10

**Authors:** Hamizah Suhaimi, Rikito Hiramatsu, Jian Xu, Tatsuya Kato, Enoch Y. Park

**Affiliations:** 1Laboratory of Biotechnology, Department of Bioscience, Graduate School of Science and Technology, Shizuoka University, 836 Ohya, Suruga-ku, Shizuoka 422-8529, Japan; noor.hamizah.binsuhaimi.16@shizuoka.ac.jp (H.S.); kato.tatsuya@shizuoka.ac.jp (T.K.); 2Laboratory of Biotechnology, Department of Applied Biological Chemistry, College of Agriculture, Graduate School of Integrated Science and Technology, Shizuoka University, 836 Ohya Suruga-ku, Shizuoka 422-8529, Japan; hiramatsu.rikito.17@shizuoka.ac.jp; 3Laboratory of Biotechnology, Research Institute of Green Science and Technology, Shizuoka University, 836 Ohya Suruga-ku, Shizuoka 422-8529, Japan; xu.jian@shizuoka.ac.jp

**Keywords:** BmNPV bacmid, nanobiomaterials, *Neospora caninum*, *Neospora caninum* profilin, neosporosis, silkworm expression system

## Abstract

Neosporosis, which is caused by *Neospora caninum*, is a well-known disease in the veterinary field. Infections in pregnant cattle lead to abortion via transplacental (congenitally from mother to fetus) transmission. In this study, a *N. caninum* profilin (NcPROF), was expressed in silkworm larvae by recombinant *Bombyx mori* nucleopolyhedrovirus (BmNPV) bacmid and was purified from the hemolymph. Three NcPROF constructs were investigated, native NcPROF fused with an N-terminal PA tag (PA-NcPROF), PA-NcPROF fused with the signal sequence of bombyxin from *B. mori* (bx-PA-NcPROF), and bx-PA-NcPROF with additional C-terminal transmembrane and cytoplasmic domains of GP64 from BmNPV (bx-PA-NcPROF-GP64TM). All recombinant proteins were observed extra- and intracellularly in cultured Bm5 cells and silkworm larvae. The bx-PA-NcPROF-GP64TM was partly abnormally secreted, even though it has the transmembrane domain, and only it was pelleted by ultracentrifugation, but PA-NcPROF and bx-PA-NcPROF were not. Additionally, bx-PA-NcPROF-GP64TM was successfully purified from silkworm hemolymph by anti-PA agarose beads while PA-NcPROF and bx-PA-NcPROF were not. The purified bx-PA-NcPROF-GP64TM protein bound to its receptor, mouse Toll-like receptor 11 (TLR-11), and formed unique nanoparticles. These results suggest that profilin fused with GP64TM was secreted as a nanoparticle with binding affinity to its receptor and this nanoparticle formation is advantageous for the development of vaccines to *N. caninum*.

## 1. Introduction

Neosporosis is a disease caused by apicomplexan parasites such as *Neospora caninum* and *Neospora hugheshi* [1]. Considering its morphology, *N. caninum* is quite similar to *Toxoplasma gondii*, so neosporosis was misdiagnosed as *T. gondii* infection until 1988 [2]. Although the morphologies are quite similar, these parasites are biologically different and are considered as zoonotic compared with toxoplasmosis where it affects human, sheep and many other warm-blooded animals [3]. Neosporosis infection occurs from unsporulated oocysts in feces followed by ingestion by the cattle, transformation from the tachyzoites stage to the sporozoites stage and leading to neosporosis in the cattle [4]. Recrudescence of the infection occurs in pregnant cattle, where this parasite is transmitted from the placenta to the unborn fetus and may lead to abortion [5,6].

*N. caninum,* a parasite of the phylum Apicomplexa, has been known as the etiologic agent of neosporosis disease [5]. Apicomplexa contains specific organelles, micronemes, rhoptries, and dense granules. Most of the specific antigen proteins are secreted from each organelle and are located at the apical end of the *N. caninum* parasite and have been extensively studied as recombinant vaccine candidates against *N. caninum* infection [7,8]. In our previous study, we demonstrated the capability of baculoviruses displaying *N. caninum*-derived antigens, such as surface antigen 1 (NcSAG1), SAG1-related sequence 2 (NcSRS2) and microneme protein 3, as an alternative method to control neosporosis because the combination of three antigens could induce T-cell activation and interferon gamma (IFN-γ) production and suppress *N. caninum* infection in mice [9].

Profilin of *N. caninum* (NcPROF) is recognized and conserved as a vaccine candidate with high potential against neospora infections [10]. Interestingly, profilin is known as a small actin-binding protein located at the apical end of *N. caninum* tachyzoites and is essential for invasion of the host cell by regulating the polymerization and depolymerization of actin filaments [11]. Furthermore, Jenkins et al. [12] and Mansilla et al. [13] revealed that *T. gondii* profilin binds to Toll-like receptor 11 (TLR11) in mice and is responsible for activating dendritic cells and stimulating the release of cytokines such as interleukin 12 (IL-12) and IFN-γ. According to Innes et al. [14], some cytokines help in controlling the infection by inhibiting parasite multiplication.

In this study, to produce recombinant NcPROF, the *Bombyx mori* nucleopolyhedrovirus (BmNPV) bacmid-based silkworm expression system was used. The BmNPV bacmid-based silkworm expression system contributes to several advantages for recombinant protein expression, such as low cost, ease of treatment and high safety [15]. Previous studies of recombinant NcPROF have not dealt with the expression fusion of the transmembrane and cytoplasmic domains of GP64 from BmNPV. Therefore, to evaluate the expression and purification of fusion native NcPROF fused with an N-terminal PA tag, fused with the signal sequence of bombyxin from *B. mori* (bx-PA-NcPROF) fused with the transmembrane and cytoplasmic domains of GP64 from BmNPV (bx-PA-NcPROF-GP64TM), NcPROF was expressed in two other NcPROF constructs, one fused with a PA tag (PA-NcPROF) and PA-NcPROF fused with the signal sequence of bombyxin from *B. mori* (bx-PA-NcPROF). The bx-PA-NcPROF-GP64TM was successfully purified from silkworm hemolymph and the binding of bx-PA-NcPROF-GP64TM with recombinant mouse TLR11 (mTLR11), and the morphological analysis of the nanoparticles were reported.

## 2. Materials and Methods

### 2.1. Construction of the Recombinant BmNPV Bacmid Containing NcPROF Constructs

The NcPROF gene (GenBank accession no. BK006901.1) was used to express recombinant NcPROF. The bx-PA (Peptide tag, GVAMPGAEDDVV)-NcPROF-GP64TM gene was synthesized by GENEWIZ Japan (Saitama, Japan) with the bombyxin signal peptide (bx) sequence (NCBI reference sequence no. NP_001103771.1) at its N-terminus and transmembrane cytoplasmic domains of GP64 from BmNPV (GP64TM) (GenBank accession no. BAF32568.1) at its C-terminus. The synthetic sequence of bx-PA-NcPROF-GP64TM was digested using *Eco*RI (NEB, Tokyo, Japan) and *Not*I (NEB, Tokyo, Japan) and was cloned into pFastBac1 (ThermoFisher Scientific. K.K., Tokyo, Japan), resulting in the construct pFast/bx-PA-NcPROF-GP64TM. The gene coding bx-PA-NcPROF was amplified by polymerase chain reaction (PCR) using (bx-PA-NcPROF) with primer sets (Table 1). The amplified gene was digested by *Eco*RI and *Xho*I and was cloned into pFastbac1, resulting in the construct pFast/bx-PA-NcPROF. Additionally, pFast/PA-NcPROF and pFast/bx-PA-NcPROF were generated by PCR amplification with primers listed in Table 1 and then underwent self-ligation. Subsequently, the constructed pFast/bx-PA-NcPROF-GP64TM, pFast/bx-PA-NcPROF, and pFast/bx-PA-NcPROF were transformed into *Escherichia coli* BmDH10Bac, respectively, as described previously [15]. The white colonies were identified as positive transformants containing each recombinant BmNPV bacmid.

### 2.2. Expression of Recombinant NcPROF in Silkworm Larvae and Bm5 Cells

Each recombinant BmNPV bacmid DNA (10 or 20 μg) was mixed with 0.1% chitosan (ratio of amino group to phosphate group 2) and 2% (*w*/*v*) of 2-(*N*-morpholino) ethanesulfonic acid (MES) buffer [16]. The mixture of each recombinant BmNPV bacmid DNA was incubated at room temperature (RT) for approximately 45 min before injection into the silkworm larvae. Subsequently, approximately 50 μL of the mixture was injected into a fifth instar silkworm larva (Ehime Sansyu, Ehime, Japan), and the larvae were reared at 25 °C and 65 ± 5% relative humidity with the artificial diet Silkmate S2 (Nosan, Yokohama, Japan) for 6–7 day. The collected hemolymph was diluted with 10-fold of phosphate-buffered saline (PBS, pH 7.4) and was injected again into fifth-instar silkworm larva and reared for 4–5 d. The larval hemolymph was collected through cutting the caudal leg, was mixed with 5 μL of 200 mM 1-phenyl-2-thiourea, and then was centrifuged at 10,000× *g* for 10 min at 4 °C. The larval fat body was dissected in PBS mixed with 0.1% Triton-X 100 before sonication on ice with an interval time of 15 s until the solution was clear and then it was centrifuged at 10,000× *g* for 30 min at 4 °C. For the expression of NcPROF in Bm5 cells, diluted hemolymph containing recombinant BmNPVs was added to the cultured Bm5 cells in a 6-well plate at 27 °C in Sf-900II medium (ThermoFisher Scientific K.K.) supplemented with 1% antibiotic-antimycotic (ThermoFisher Scientific K.K.) and 10% FBS (Gibco, Tokyo, Japan). The culture supernatant and infected-Bm5 cells were collected at 3 d post infection. The culture supernatant (culture media) was collected after centrifugation at 10,000× *g* for 10 min at 4 °C together with the same solution as the larval hemolymph. Infected-Bm5 cells were mixed with the same solution and were subjected to the same treatment as the larval fat body, followed by separation of cell lysate soluble and insoluble fractions after centrifugation. The supernatants of the larval hemolymph, fat body, cell medium, cell lysate soluble, and insoluble were immediately frozen at −80 °C until further analysis.

### 2.3. Purification of bx-PA-NcPROF-GP64TM from Silkworm Larvae

bx-PA-NcPROF-GP64TM was purified from silkworm larval hemolymph using anti-PA tag affinity chromatography (WAKO Pure Chemical Industries, Osaka, Japan). Ten-times the diluted supernatant of the hemolymph was mixed with anti-PA tag affinity beads and incubated at 4 °C for 24 h with gentle agitation. Next, the beads were washed 10 times with Tris-buffered saline (TBS, pH 7.6), and the bound proteins were eluted with 0.1 M glycine-HCl (pH 3) and immediately neutralized with 1.5 M Tris-HCl (pH 8.0). The elution was conducted at RT in a stepwise manner.

### 2.4. Ultracentrifugation Analysis of bx-PA-NcPROF-GP64TM, bx-PA-NcPROF, and PA-NcPROF

Two milliliters of crude hemolymph was mixed with 1 mL of PBS (pH 6.2). This mixture was centrifuged at 100,000× *g* for 90 min at 4 °C, and the pellet was suspended in 1 mL of PBS (pH 6.2). This suspension was sonicated to dissolve the pellet, and the suspension was collected and subjected to western blotting.

### 2.5. Transmission Electron Microscopy Observation of bx-PA-NcPROF-GP64TM

Purified bx-PA-rNcPROF-GP64TM was analyzed through negative staining. The bx-PA-rNcPROF-GP64TM (20 μL) drop was loaded onto the surface of film 200 mesh copper grid (Nisshin Em Co. Ltd., Tokyo, Japan) within 30 s at RT. Next, the grid was washed 3 times with PBS and negatively stained with phosphotungstic acid (2% of *v*/*v*). To investigate the surface of the bx-PA-rNcPROF-GP64TM nanoparticle, the sample on the grid was blocked in 2% (*v*/*v*) bovine serum albumin (BSA) for approximately 5 min after washing with PBS 3 times. Next, the grid was incubated for 1 h at RT, and then it was loaded onto the surface drop of rat anti-PA tag monoclonal antibody (NZ-1, 1:50 in PBS) (WAKO Pure Chemical Industries). After 1 h of incubation, the grid was washed 6 times and was loaded onto the surface with goat anti-rat immunoglobulin G (IgG) (H+L)-conjugated with 12 nm gold beads (1:50 in PBS) (Jackson ImmnunoResearch Inc., West Grove, PA, USA) drops and was incubated for 1 h at RT. Finally, the grid was washed 6 times with PBS, followed negative staining with phosphotungstic acid (2% *v*/*v*). Images were acquired with a transmission electron microscope (TEM; JEM-2100F; JEOL, Ltd., Tokyo, Japan) operated at 100 kV.

### 2.6. Sodium Dodecyl Sulfate-Polyacrylamide Gel Electrophoresis (SDS-PAGE) and Western Blotting

The collected larval hemolymph and extract of a fat body were used to confirm the expression of recombinant proteins through SDS-PAGE (Bio-Rad, Hercules, CA, USA). The proteins were transferred onto a polyvinylidene fluoride (PVDF) membrane using a trans-blot SD semidry transfer cell (Bio-Rad). Next, 5% (*w*/*v*) skimmed milk in Tris-buffered saline containing 0.1% (*v*/*v*) Tween 20 (TBST) was used to block the PVDF membrane for 1 h. Thereafter, the membrane was incubated with rat anti-PA tag monoclonal antibody (NZ-1, 0.1 μg/mL) (Wako Pure Chemical Industries, Ltd.), followed by incubation with horseradish peroxidase (HRP)-conjugated goat anti-rat IgG (H+L) (1:10,000) (Bios Antibodies Inc., Woburn, MA, USA) and development for 1 min with Immobilon western chemiluminescence HRP substrate (Merck Millipore, Burlington, MA, USA). Stained proteins were detected using a molecular imager VersaDoc MP imaging systems (Bio-rad).

### 2.7. Binding Assay of bx-PA-NcPROF-GP64TM with Mouse TLR11

The binding assay for purified bx-PA-NcPROF-GP64TM against recombinant mTLR11 Fc chimera (R&D Systems, Minneapolis, MN, USA) was carried out by enzyme-linked immunosorbent assay (ELISA). The binding assay was performed in 96-well plates coated with 50 µL/well of mTLR11 (100 ng/well) at 4 °C overnight. The plates were blocked with 100 µL/well of 2% skimmed milk in PBS for 2 h at RT. After 2 h, the plates were washed 3 times with PBS with 0.1% Tween 20 (PBST). The purified bx-PA-NcPROF-GP64TM with different concentrations (100, 300 and 500 ng/well) with the 1 mM dithiothreitol (DTT) treatment and 100 ng without 1 mM DTT treatment was diluted in blocking buffer in triplicate, followed by incubation for 2 h at RT. The plate was washed with PBST and incubated for 2 h at RT with 50 µL/well of rat anti-PA tag monoclonal antibody (NZ-1, 1:1000 dilution). Subsequently, the plates were washed again with PBST and were incubated for 2 h at RT with 50 µL/well of HRP-conjugated goat anti-rat IgG (H+L) secondary antibody (1:5000 dilution). Next, the plates were washed with PBST and developed by incubating with 50 µL/well of 3,3′,5,5′-tetramethylbenzidine (TMBZ) (Dojindo Co. Ltd., Kanagawa, Japan) for 20 min, followed by stopping the reaction with 100 µL/well of stop solution (10% H_2_SO_4_). Thereafter, 1 mM DTT and BSA (100 ng/well) were used as negative controls. Finally, the absorbance of the mixture was measured using a microplate reader (Model 680, Bio-Rad, Hercules, CA, USA) at 450 nm.

## 3. Results and Discussion

### 3.1. Expression of bx-PA-NcPROF-GP64TM, bx-PA-NcPROF, and PA-NcPROF in Silkworms

In this study, three NcPROF constructs were generated. PA-NcPROF has a PA tag sequence at its N-terminus (Figure 1A), whereas bx-PA-NcPROF has the signal sequence of bombyxin from *B. mori* at its N-terminus (Figure 1B). Meanwhile, bx-PA-NcPROF-GP64TM has the transmembrane and cytoplasmic domains of GP64 from BmNPV at the C-terminus of bx-PA-NcPROF (Figure 1C).

The bombyxin signal peptide (bx) is responsible for the secretion of recombinant proteins to the hemolymph of silkworm larvae [17,18]. The GP64TM were used as a fusion partner with NcPROF as a target protein to be incorporated into the cell membrane [19]. All NcPROF constructs were observed in both the fat body and hemolymph when they were expressed in silkworm larvae (Figure 2A). PA-NcPROF, bx-PA-NcPROF, and bx-PA-NcPROF-GP64TM comprise 176, 195, and 226 amino acids, respectively, and their estimated molecular weights are 19, 21, and 25 kDa, respectively. All NcPROFs were expressed and verified intra- and extracellularly in Bm5 cells (Figure 2B). The molecular weight of PA-NcPROF was approximately 25 kDa by western blotting.

The molecular weight of native NcPROF in this study was similar to that of full-length profilin expressed in *E. coli* (22 kDa) [12]. This finding suggests that PA-NcPROF expressed in silkworm larvae may be not modified posttranslationally. bx-PA-NcPROF was observed in the fat body and hemolymph at 25–30 kDa as a large obscure band that appeared as two bands. The molecular weight of the lower band was the same as that of PA-NcPROF and that of the upper band was larger than that of PA-NcPROF. This finding suggests that bx-PA-NcPROF may be partially modified posttranslationally after the cleavage of the bombyxin signal peptide. In fact, N128 was predicted as an *N*-glycosylation site estimated by NetNGlyc 1.0 Server (http://www.cbs.dtu.dk/services/NetNGlyc/). The bx-PA-NcPROF-GP64TM product was observed at 30–40 kDa by western blotting. Interestingly, bx-PA-NcPROF-GP64TM was observed in the hemolymph and culture medium although it was fused with the transmembrane domain of GP64. In our previous report, *N. caninum*-derived antigens NcSAG1 and NcSRS2 fused with the transmembrane and cytoplasmic domains of GP64 were not observed in hemolymph [9]. In this study, bx-PA-NcPROF-GP64TM was secreted into hemolymph, in part, and the culture medium from Bm5 cells. It has been reported that profilins of *T. gondii* and *Babesia canis* were partially secreted even though no signal peptide is predicted in these genes [20,21]. In this study, NcPROF was also secreted regardless of the addition of the signal sequence. To investigate *N*-glycosylation of the *N*-glycan of three NcPROFs, PNGase F treatment was carried out (Appendix A). Molecular weight of each NcPROF was not changed between before and after the PNGase F treatment, indicating no *N*-glycan was attached into the three NcPROFs. These results suggest that the two bands of bx-PA-NcPROF did not come from its *N*-glycosylation. Hence, it showed that the bx-PA-NcPROF and bx-PA-NcPROF-GP64TM were not posttranslationally modified with an *N*-glycan in endoplasmic reticulum even though these proteins have the bx signal peptide. In addition, the discrepancy of the molecular weight of each PROF to its estimated that was not caused by the *N*-glycosylation. In nature, estimated pI of this NcPROF is around 4.89, which may have influence on the mobility of NcPROFs on the SDS-PAGE gel [22]. In addition, the negative charge of acidic residues may create repulsion and this repulsion caused the anomalous migration of protein in SDS-polyacrylamide gels [23]. This may be the reason why the molecular weight of bx-PA-NcPROF-GP64TM was increased.

The expressed NcPROFs in the Bm5 cell culture broth were further investigated by ultracentrifugation (Figure 3). It showed that PA-NcPROF and bx-PA-NcPROF were not precipitated by ultracentrifugation (100,000× *g*), while bx-PA-NcPROF-GP64TM was pelleted. Normally, transmembrane proteins are anchored in the membrane fraction of cells and are not easily secreted. However, glycoproteins from some viruses are secreted and are partially incorporated into extracellular vesicles when they are overexpressed [24,25]. Moreover, the transmembrane and cytoplasmic domains of GP64 from BmNPV facilitated the display of recombinant proteins on BmNPV particles. These findings suggest that bx-PA-NcPROF-GP64TM was secreted as nanoparticles or by display on the envelope of BmNPV particles.

bx-PA-NcPROF-GP64TM was then purified from silkworm hemolymph using anti-PA tag agarose. As shown in Figure 4A, bx-PA-NcPROF-GP64TM was successfully purified as a single band in SDS-PAGE and was further verified by western blotting (Figure 4B). This finding indicates that bx-PA-NcPROF-GP64TM does not display on the surface of BmNPV particles because no band was observed in the purified sample except for bx-PA-NcPROF-GP64TM. These results suggested that bx-PA-NcPROF-GP64TM forms only some nanoparticles and can be secreted into hemolymph and the culture broth. The yield of purified NcPROF-GP64TM was approximately 14 µg from 3 mL of larval hemolymph.

Interestingly, PA-NcPROF and bx-PA-NcPROF could not be purified from hemolymph using the same protocols as bx-PA-NcPROF-GP64TM. From the results, it was demonstrated that, without modification of the transmembrane region at the C-terminus for both constructs, bx-PA-NcPROF was only slightly purified and PA-NcPROF was almost impossible to be purified from hemolymph (Appendix A) although a portion of bx-PA-NcPROF was secreted into hemolymph compared with bx-PA-NcPROF-GP64TM (Figure 2A). However, when 0.5% (*w*/*v*) nonyl phenoxypolyethoxylethanol (NP-40) was added, both bx-PA-NcPROF and PA-NcPROF could be purified from hemolymph and showed positive bands at 30 kDa and 25 kDa, respectively (Appendix A), suggesting that bx-PA-NcPROF and PA-NcPROF were aggregated abnormally and the aggregates were dissolved with NP-40.

### 3.2. Binding Assay of bx-NcPROF-GP64TM with mTLR11

Recently, it was reported that TLR is responsible for parasite recognition and the induction of cytokines in *T. gondii* [26]. NcPROF induces limited protection and the T-cell response in mice [13]. In this study, NcPROF expressed in silkworms was investigated to determine whether it binds to recombinant mTLR11. The binding of purified bx-NcPROF-GP64TM to mTLR11 was not clearly observed compared with that of the negative control. Because bx-PA-NcPROF-GP64TM formed multimers under nonreducing conditions, DTT was then added to purified bx-PA-NcPROF-GP64TM before ELISA (Figure 5A,B). As shown in Figure 5C, the specific binding was enhanced by an increment of almost 72.7% after DTT treatment and simultaneously increased the binding ability when different concentrations (100, 300, 500 ng/well) were challenged. Hence, it suggests that bx-PA-NcPROF-GP64TM expressed in silkworm larvae is functional. These results indicate that multimer formation might prevent bx-PA-NcPROF-GP64TM from binding to mTLR11. Similar to the results of a previous study, Hedhli et al. [27], showed that PROF of *T. gondii* secreted into *Drosophila* S2 cell broth with the signal sequence of insect binding immunoglobulin protein (BiP) enhanced the cellular and humoral responses in mice after DTT treatment. These results indicate that bx-PA-NcPROF-GP64TM was secreted through the secretory pathway in the host and irregular disulfide bonds were formed in the endoplasmic reticulum. In nature, NcPROF does not have the signal peptide at its N-terminus and does not enter the endoplasmic reticulum [20,28].

### 3.3. Morphology of bx-PA-NcPROF-GP64TM Nanoparticles

The morphology of purified bx-PA-NcPROF-GP64TM was observed using TEM. Nanoparticles with a diameter of 30 nm were observed in purified bx-PA-NcPROF-GP64TM (Figure 6A). To further analyze the purified bx-PA-NcPROF-GP64TM, immuno-TEM was conducted. The recombinant proteins were probed with rat anti-PA tag (NZ-1) and goat anti-rat IgG (H+L) conjugated with 12-nm gold beads. As shown in Figure 6B, bx-PA-NcPROF-GP64TM demonstrated a spherical morphology with a diameter of approximately 30 nm. These results indicate that bx-PA-NcPROF-GP64TM was secreted as nanoparticles (Figure 3).

Vesicular stomatitis virus G glycoprotein (VSV-G) could be recovered from the culture supernatant when VSV-G alone was expressed in mammalian cells [24,25]. Similarly, the S protein of severe acute respiratory syndrome-associated coronavirus was incorporated into exosomes when its transmembrane and cytoplasmic domains were replaced with those of VSV-G [29]. Extracellular vesicles containing virus-encoded glycoprotein were secreted into the culture supernatant when mammalian cells were infected with the modified *Vaccinia* virus Ankara strain [30]. Additionally, glycoprotein 64 (AcGP64) of *Autographa californica* multiple nucleopolyhedrovirus was secreted into the culture medium when AcGP64 was expressed in mammalian cells [31]. In this study, it is possible that the bx-PA-NcPROF-GP64TM protein formed nanoparticles when it was expressed in silkworm larvae and Bm5 cells. In our previous paper, antigens of *N. caninum* fused with the transmembrane and cytoplasmic domains of BmGP64 were displayed on the surface of BmNPV particles, but few of these fusion proteins were secreted into silkworm hemolymph [9]. These studies suggest that the secretion of recombinant fusion proteins with the transmembrane and cytoplasmic domains of BmGP64 depends on the properties of recombinant proteins of interest. Currently, nanoparticles were only observed in purified bx-PA-NcPROF-GP64TM but not in the other two constructs. Additionally, when purified bx-PA-NcPROF-GP64TM was treated with 1% TritonX-100, some of this protein moved to the fraction at low sucrose concentration during sucrose density gradient centrifugation (Appendix A), implying that this fusion protein forms nanoparticles. We are investigating whether these nanoparticles are aggregates or not and how these nanoparticles form.

Apicomplexan profilin, including NcPROF, is a promising protein target with an adjuvant activity as a vaccine candidate [10]. Previous research from Gause et al. [32], stated that delivery of the subunit antigen using particle-based delivery systems can lead to significant improvement in immunogenicity because these systems have now been enhanced by engineering through the physiochemical properties of the particle to promote an immune response. Hence, it was advantageous to purify bx-PA-NcPROF-GP64TM because it naturally forms unique functional nanoparticles without modification through its physiochemical properties to deliver subunit antigens through a particle-based delivery system. Antigen-displaying nanoparticles have been normally prepared by the co-expression of viral structural proteins and antigens or chemical conjugation and genetic fusion of antigens to virus-like particles [33,34]. In this study, NcPROF-displaying nanoparticles were prepared in insect cells and silkworm larvae only by expression of bx-PA-NcPROF-GP64TM, which has the bombyxin signal sequence and transmembrane and cytoplasmic domains of BmGP64 at its N-terminus and C-terminus, respectively.

## 4. Conclusions

bx-PA-NcPROF-GP64TM was secreted partly into the silkworm larval hemolymph and culture broth of Bm5 cells, although this protein has the transmembrane domain of BmGP64. NcPROF was purified from silkworm hemolymph as a single band only when the bombyxin signal sequence and transmembrane and cytoplasmic domains of BmGP64 were fused with the N- and C-termini, respectively. The purified bx-PA-NcPROF-GP64TM formed unique nanoparticles and was bound to mTLR11.

## Figures and Tables

**Figure 1 nanomaterials-09-00593-f001:**
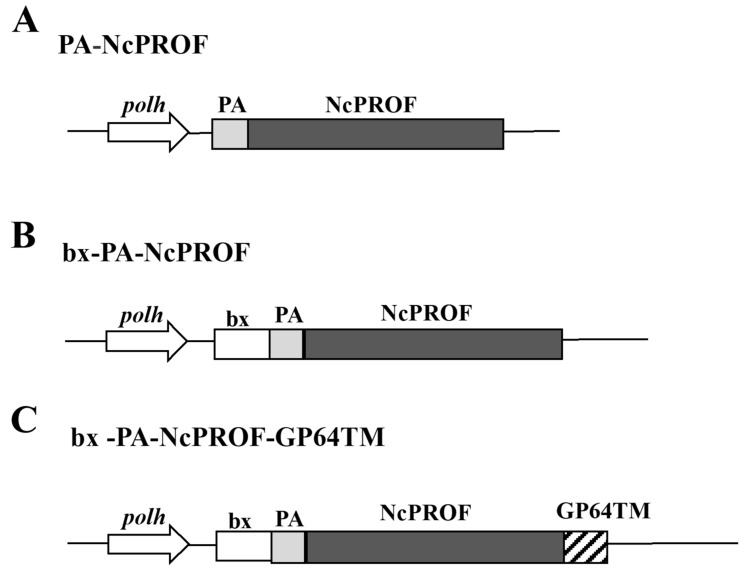
Constructs of native NcPROF fused with an N-terminal PA tag (PA-NcPROF) (**A**), PA-NcPROF fused with the signal sequence of bombyxin from *B. mori* (bx-PA-NcPROF) (**B**) and bx-PA-NcPROF with additional C-terminal transmembrane and cytoplasmic domains of GP64 from *Bombyx mori* nucleopolyhedrovirus (BmNPV) (bx-PA-NcPROF-GP64TM) (**C**). *polh*, polyhedrin promoter; bx, bombyxin signal sequence; GP64TM, transmembrane and cytoplasmic domains of GP64 from BmNPV; PA, PA-tag sequence.

**Figure 2 nanomaterials-09-00593-f002:**
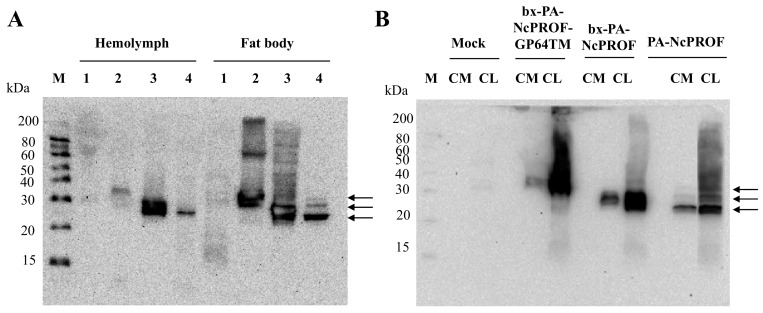
Western blot analysis of bx-NcPROF-GP64TM, bx-NcPROF and NcPROF expressed in silkworm larvae (**A**) and Bm5 cells (**B**). Recombinant BmNPV/bx-PA-NcPROF-GP64TM, BmNPV/bx-PA-NcPROF or BmNPV/PA-NcPROF were injected into the silkworms, followed by harvesting at 5 d post injection. Bm5 cells were harvested at 3 d post infection of the indicated recombinant BmNPV. Lanes M, 1, 2, 3 and 4 in (**A**) denote the molecular marker, mock, bx-PA-NcPROF-GP64TM, bx-PA-NcPROF, and PA-NcPROF, respectively. Lanes M, CM and CL in (**B**) denote the molecular marker, protein samples from cell medium and cell lysate, respectively. Upper, middle and lower arrows in (**A**,**B**) indicate bx-NcPROF-GP64TM, bx-NcPROF and NcPROF, respectively.

**Figure 3 nanomaterials-09-00593-f003:**
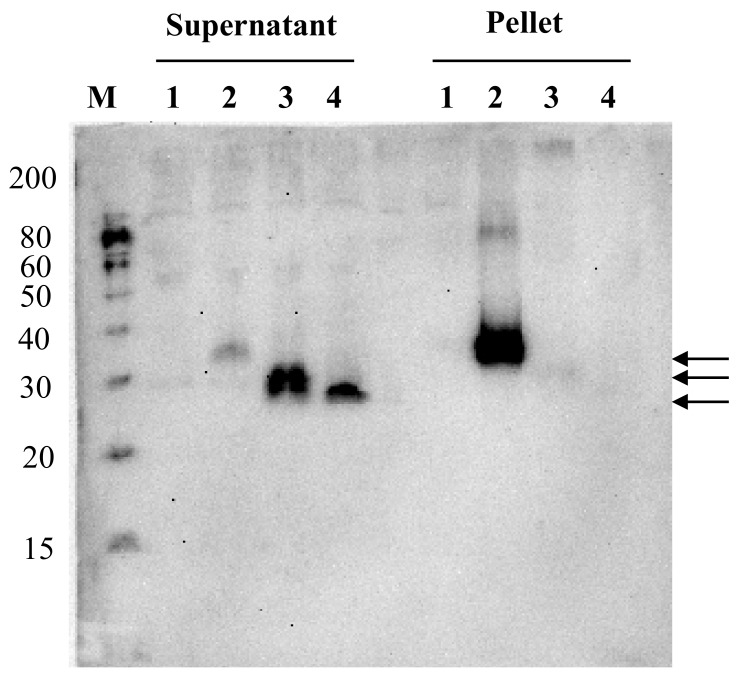
Western blots of soluble and insoluble fractions from the Bm5 cell culture supernatant. The culture supernatant containing bx-PA-NcPROF-GP64TM, bx-PA-NcPROF or PA-NcPROF was centrifuged at 100,000× *g*, and the pellet was resuspended in 1 mL of PBS. Lanes M, 1, 2, 3 and 4 denote the molecular marker, mock, bx-PA-NcPROF-GP64TM, bx-PA-NcPROF and PA-NcPROF, respectively. Upper, middle and lower arrows indicate bx-NcPROF-GP64TM, bx-NcPROF and NcPROF, respectively.

**Figure 4 nanomaterials-09-00593-f004:**
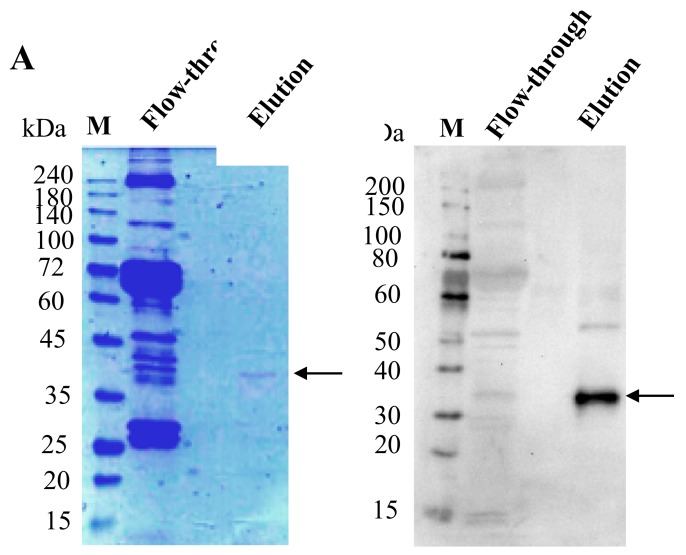
(**A**) Sodium dodecyl sulfate-polyacrylamide gel electrophoresis (SDS-PAGE) and western blot under reducing conditions. (**B**) SDS-PAGE of purified bx-PA-NcPROF-GP64TM from hemolymph using anti-PA tag affinity chromatography. M indicates the molecular weight marker; arrows indicate the recombinant bx-PA-NcPROF-GP64TM band.

**Figure 5 nanomaterials-09-00593-f005:**
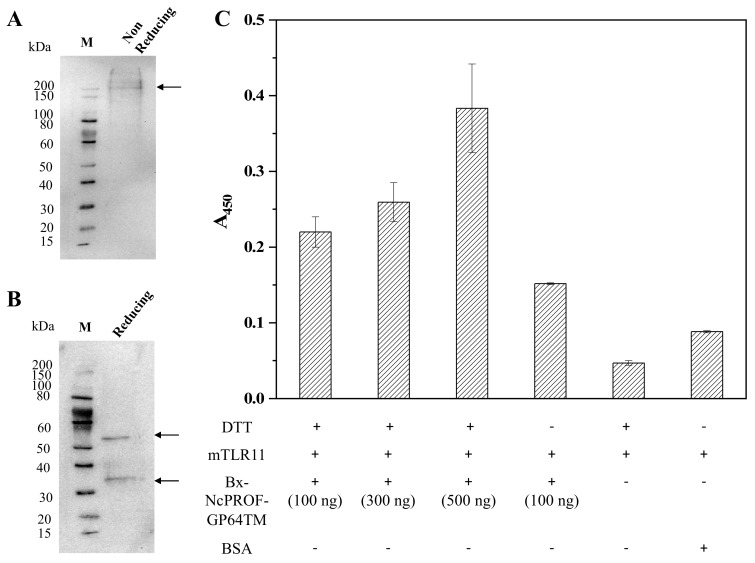
Western blots under reducing conditions (**A**) and nonreducing conditions (**B**) of purified bx-PA-NcPROF-GP64TM. (**C**) enzyme-linked immunosorbent assay (ELISA) analysis of the binding of purified of bx-NcPROF-GP64TM to recombinant mTLR11 Fc chimera. Purified bx-NcPROF-GP64TM (100 ng, 300 ng, 500 ng) with the 1 mM dithiothreitol (DTT) treatment and 100 ng of bx-NcPROF-GP64T without the 1 mM DTT treatment were assayed. Bovine serum albumin (BSA) was used as a negative control. The absorbance was detected at 450 nm and presented as means ±0.001~0.05 standard deviation with coefficient of variance ±0.7%~15%.

**Figure 6 nanomaterials-09-00593-f006:**
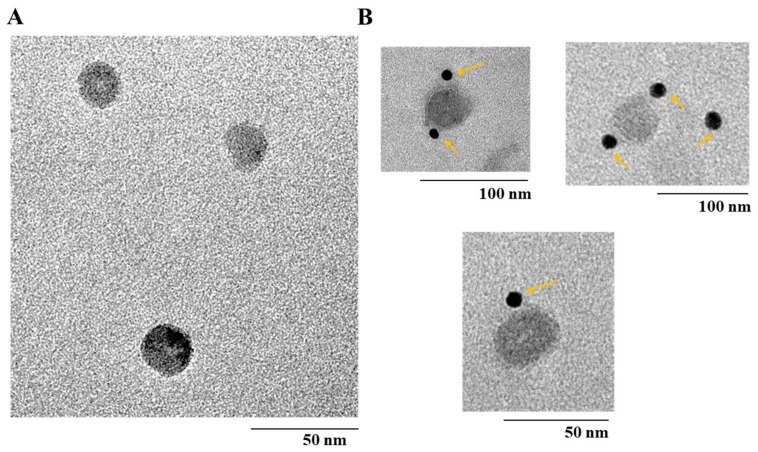
(**A**) Transmission electron microscope (TEM) image of bx-PA-NcPROF-GP64TM. (**B**) Immuno-TEM using gold-labeled antibody. The arrows in (**B**) anti-PA tag-conjugated-gold nanoparticles, respectively.

**Table 1 nanomaterials-09-00593-t001:** Primers used in this study.

Name	5′ to 3′
pFastBac1	
Forward	5′-TATTCCGGATTATTCATACC-3′
Reverse	5′-ACAAATGTGGTATGGCTGATT-3′
NcPROF	
Forward	5′-GGACACAATCGGAGAGGACG-3′
Reverse	5′-GTGCACACATGGTGATGTCG-3′
pUC/M13	
Forward	5′-CCCAGTCACGACGTTGTAAAACG-3′
Reverse	5′-AGCGGATAACAATTTCACACAGG-3′
PA-NcPROF	
Forward	5′-GGCGTTGCCATGCCAGGTGC-3′
Reverse	5′-CATGAATTCCGCGCGCTTCG-3′
Bx-PA-NcPROF	
Forward	5′-GCGCGGAATTCATGAAGATACTCCTTGCT-3′
Reverse	5′-GCATGCCTCGAGTTAATAGCCAGACTGGTGAAGGTACTCG-3′

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
