# Peer review of "Secretory Nanoparticles of Neospora caninum Profilin-Fused with the Transmembrane Domain of GP64 from Silkworm Hemolymph"

_nanomaterials, 2019, doi:10.3390/nano9040593_

Reviewer 1 Report

The authors should explain more on what GP64 is why it is important to evaluate its fusions with profilin.

The data on the TLR11 in Figure4 are weak, the OD values are rather low in comparison with the negative control ; the authors should do a dose response  and inhibition experiments using a known agonist/ antagonist for TLR11 to demonstrate the binding is real.

In Figure 6 a negative control should be included. In 6A the background in TEM is very high, difficult to see the gold particles.

Author Response

Dear Reviewer #1, 

 Please find the enclosed revised version of the manuscript, “Secretory nanoparticles of Neospora caninum profilin fused with the transmembrane domain of GP64 from silkworm hemolymph” Manuscript ID: nanomaterials-459171by Suhaimi et al. and we are submitting revised MS for the publication in Nanomaterials as follow, 

 Reviewer #1:

1)    The authors should explain more on what GP64 is why it is important to evaluate its fusions with profilin.

Response: Referring to L 203-204 of old MS we explained the GP64 and its function together with reference citation. Furthermore, the reason why the GP64 is important to evaluate the fusions with profilin was explained in L231–239 of revised MS.

    L230-238

“Interestingly, bx-PA-NcPROF-GP64TM was observed in the hemolymph and culture medium   although it was fused with the transmembrane domain of GP64. In our previous report, N. caninum-derived antigensNcSAG1 and NcSRS2 fused with the transmembrane and cytoplasmic domains of GP64 were not observed in hemolymph [9]. In this study, bx-PA-NcPROF-GP64TM was secreted into hemolymph, in part, and the culture medium from Bm5 cells. It has been reported that profilins of T. gondiiand Babesia caniswere partially secreted even though no signal peptide is predicted in these genes [20, 21]. In this study, NcPROF was also secreted regardless of the addition of the signal sequence.”

2)    The data on the TLR11 in Figure4 are weak, the OD values are rather low in comparison with the negative control; the authors should do a dose response and inhibition experiments using a known agonist/ antagonist for TLR11 to demonstrate the binding is real.

Response: Thank for valuable comments, we added new experimental results. We challenged the concentration of bx-NcPROF-GP64TM from 300 to 500 ng in the presence of DTT. The binding ability increased ~72.7% with the increase of purified bx-NcPROF-GP64TM from 100 ng to 500 ng (revised Figure 5C). It strongly supports the binding ability with bx-NcPROF-GP64TM and we described this result in revised Figure 5C (L315–320) of revised MS.

 3)     In Figure 6 a negative control should be included. In 6A the background in TEM is very high, difficult to see the gold particles. 

Response: We are sorry for a little noisy image. We replaced the Figure 6A with the clear TEM image (attached below). In conclusion, we observed particles of purified bx-NcPROF-GP64TM from silkworm larval hemolymph as shown in Fig. 6A of revised MS. Whereas, the Fig. 6B of revised MS was immuno-TEM, where the antibodies goat anti-rat IgG (H+L) conjugated with the 12 nm gold beads were bound the particles. This result shows that the bx-NcPROF-GP64TM was secreted and formed the particles with diameter of approximately 30 nm.

We revised old MS and submitted revised MS with track changeable version for reviewers’ convenience. 

Reviewer 2 Report

In this manuscript, the authors found the purified bx-PA-NcPROF-GP64TM protein binds to its receptor, mouse TLR-11 and the nanoparticle formation from the profilin fused with GP64TM is potentially used for the development of vaccines to Neospora caninum. The N- and C-termini are important domains to have NcPROF-displaying nanoparticles. Overall, the topic of the manuscript is appropriate to the review for Nanomaterials and the manuscript is well written.  Considering that Nanomaterials publishes considerably novel studies and findings in the synthesis and use of nanomaterials, I would recommend the publication of this manuscript on Nanomaterials. However, there are minor changes I would recommend and the following items need to be corrected in a revision:

 In page 6, lines 216-219, suggest rephrasing the statements, to avoid using “These(or This) result(s)” repeatedly.

 In page 6, line 221, what does ‘some influence’ mean? The authors should explain more details using proper references.

 In page 8, line 276, suggest citing some references to support the statement, unless the experimental data exists.

Author Response

Dear Reviewer #2, 

Please find the enclosed revised version of the manuscript, “Secretory nanoparticles of Neospora caninum profilin fused with the transmembrane domain of GP64 from silkworm hemolymph” Manuscript ID: nanomaterials-459171by Suhaimi et al. and we are submitting revised MS for the publication in Nanomaterials as follow, 

Reviewer #2: 

In this manuscript, the authors found the purified bx-PA-NcPROF-GP64TM protein binds to its receptor, mouse TLR-11 and the nanoparticle formation from the profilin fused with GP64TM is potentially used for the development of vaccines to Neospora caninum. The N- and C-termini are important domains to have NcPROF-displaying nanoparticles. Overall, the topic of the manuscript is appropriate to the review for Nanomaterials and the manuscript is well written.  Considering that Nanomaterials publishes considerably novel studies and findings in the synthesis and use of nanomaterials, I would recommend the publication of this manuscript on Nanomaterials. However, there are minor changes I would recommend and the following items need to be corrected in a revision:

1)    In page 6, lines 216-219, suggest rephrasing the statements, to avoid using “These(or This) result(s)” repeatedly.

              Response: Thank you very much for the suggestion. We corrected as follow, 

L242–245 of revised MS

These results suggest that the two bands of bx-PA-NcPROF did not come from its N-glycosylation. Hence, it showed that the bx-PA-NcPROF and bx-PA-NcPROF-GP64TM were not posttranslationally modified with an N-glycan in endoplasmic reticulum even though these proteins have the bx signal peptide. ‘’

 2)    In page 6, line 221, what does ‘some influence’ mean? The authors should explain more details using proper references.

Response: We added more details of “some influence” in revise MS with several new references (L246–251 of revised MS).

‘’ In nature, estimated pI of this NcPROF is around 4.89, which may have influence on the mobility of NcPROFs on the SDS-PAGE gel [22]. In addition, the negative charge of acidic residues may create repulsion and this repulsion caused the anomalous migration of protein in SDS-polyacrylamide gels [23]. This may be the reason why the molecular weight of bx-PA-NcPROF-GP64TM was increased. ‘’

3) In page 8, line 276, suggest citing some references to support the statement, unless the experimental data exists.

Response: We cited references as the reviewer’s indication (L310–311 of revised MS). 

“In nature, NcPROF does not have the signal peptide at its N-terminus and does not enter the endoplasmic reticulum [20, 28].” 

We revised old MS and submitted revised MS with track changeable version for reviewers’ convenience. 

Round  2

Reviewer 1 Report

I am happy with the response to my comments.

This manuscript is a resubmission of an earlier submission. The following is a list of the peer review reports and author responses from that submission.

Round  1

Reviewer 1 Report

this work is in the field of vaccination and biology with no relation to nanomedicine, although there are nanoparticles that are formed in the process but this does not justify publication in a nano-journal. The article is suited for publication in a relevant journal in the field of biology and vaccination, there is no interest to the nano-community. 

Reviewer 2 Report

The manuscript is very clear and relevant to scientific community.

Please revise the english language and uniformize the writing in the manuscript, for example line 135 is written "sec" instead of "s" as in the remaining text. 

Reviewer 3 Report

This paper deals with nanoparticles/exosomes produced by silkworm larvae or Bm5 cells infected with a vector for the expression of Profilin from Neospora caninum. Three different constructs were used and the resulting expressed proteins were analysed.

The authors are not expert in immunology, as can be seen from the way they use words of this discipline. Indeed, the title of the manuscript (Title: Secretory nanoparticles of Neospora caninum profilin fused with the transmembrane domain of GP64 from silkworm hemolymph targeting a vaccine against neosporosis) presented by Hamizah Suhaimi et al., is not appropriate and misleading, since they do not show any results of vaccination.

In line with the above, the innate immune protein (line 21) does not mean anything to me. A protein can mediate/be responsible for an innate immune response, but as written on line 62 the authors report that the protein the “response elicited by immunization in mice with …”. Again the word “regulatory” does not means anything.

Line 309: the adjuvant activity is not within the immunizing protein. Again no clear idea of immunology.

So, more care should be used when writing concepts.

Line 93: E. coli are transformed by the constructs, but these are not transformed into E. coli. Again, the language is not appropriate.

It is not clear from where this GP64TM comes. Only at the end of the manuscript I understand that it could be a protein of the silkworm. Here is some confusion about this fragment of protein: in Fig 1 it is sketched as GP64TM, but along in the manuscript the authors speak about transmembrane and cytoplasmic domain. This must be clarified.

Paragraph from line 182 to line 190: the estimated molecular weight do not correspond to what found (they say 25 kDa). The authors speak about the possibility of a glycosylation of the molecule. They should stress more, if there are data from the literature.

Fig. 2B. I suggest to eliminate the two lanes at extreme right CLS and CLI, and replace them with only CL, since they are reduntant. Indeed these results are presented also in Fig. 3.

Fig. 4. The major bands marked with arrows do not display the same molecular weight. Maybe that only the one of apparent lower molecular weight in Fig. 4A is corresponding to the one in Fig. 4B. This may affect the calculations of the yields of the protein.

The major point is about the data and the conclusions about aggregation of the two lower molecular weight proteins (without GP64) and the ability to pellet the one with GP64. Since the former are devoid of a transmembrane segment they should be pelleted more easily than the latter, which should be associated with some phospholipids of the cell membrane (indeed the authors show that are nanoparticles) and thus more difficult to be pelleted.

When speaking about these nanoparticles/exosomes the authors should clearly say that they are secreted and not that they are extracellular. By definition secretion is in the extracellular compartment. This is pleonastic.

Fig. 6: TEM. There is not such a big difference in the size of the exosome, since the three bars are referred to different sizes. It can be easily concluded that exosomes are around 50 nm.

Finally, the authors conclude “NcPROF was purified from silkworm hemolymph as a single band only when the bombyxin  signal sequence and transmembrane  and  cytoplasmic  domains  of  BmGP64  were  fused  with  the  N -  and  C-termini,  respectively.” Since from the manuscript it is clear that similar proteins could be secreted even in the absence of a leader sequence, the authors should test also a PA-NcPROF-GP64TM construct without bx leader. I guess that this could perform as well as the bx- PA-NcPROF-GP64TM construct.

 AT the end: abbreviations: I would add PA …

 All these comments show that the manuscript is not ready for publication in this form.